# The Leaf Color and Trichome Density Influence the Whitefly Infestation in Different Cassava Cultivars

**DOI:** 10.3390/insects14010004

**Published:** 2022-12-21

**Authors:** Marcelo A. Pastório, Adriano T. Hoshino, Cíntia S. G. Kitzberger, Orcial C. Bortolotto, Luciano M. de Oliveira, Adevanir Martins dos Santos, Wilmar F. Lima, Ayres de O. Menezes Junior, Humberto G. Androcioli

**Affiliations:** 1Department of Entomology, Universidade Estadual do Oeste do Paraná, Marechal Cândido Rondon, Cascavel 85819-110, Brazil; 2Department of Agronomy, Universidade Estadual de Londrina, Londrina 86057-970, Brazil; 3Instituto de Desenvolvimento Rural do Paraná IAPAR-EMATER, Curitiba 86047-902, Brazil; 4Department of Agronomy, Universidade Estadual do Centro-Oeste, Guarapuava 85040-080, Brazil

**Keywords:** *Manihot esculenta*, *Bemisia tuberculata*, leaf colorimetric parameters, leaf pubenscence

## Abstract

**Simple Summary:**

The cassava is an important food crop in many regions of the world, mainly in the African continent and Brazilian regions. Pest occurrence has been the main limiting factor towards high productivity. The objective of this study was to identify the relationship between two whitefly species’ occurrences and leaf characteristics in different cassava cultivars. Fewer whitefly nymphs were observed in cultivars with the lowest trichome density, highest light reflection and chroma in leaves. These cultivars could be recommended in areas where whiteflies have an infestation history. The morphological characteristics of cassava leaves such as trichome density, light reflection and chroma, should all be considered in future resistance breeding programs.

**Abstract:**

The whitefly species *Bemisia tuberculata* and *Aleurotrixus aepim* (Hemiptera: Aleyrodidae) are considered important cassava (*Manihot esculenta*) pests. Leaf color and other morphological characteristics can influence the pest’s interactions with the host plants. Thus, this study aimed to identify the relationship between whitefly occurrence and trichome density and leaf color in different cassava cultivars. The study was conducted in the field during the 2014/2015 and 2016/2017 crop seasons. The whitefly occurrence was surveyed in the cultivars, IAPAR 19, IPR Upira, IPR União, IAC 576-70, IAC 14, IAC 90, Catarina Branca, Santa Helena and Baianinha. The whitefly nymph quantification was correlated with non-glandular trichome density, luminosity (L*) and chroma (a* and b*) of the cassava leaves. IAPAR 19 and IAC 14 were less infested by whitefly nymphs when contrasted with IPR União, IPR Upira and Baianinha, which were the most infested. The lowest *B. tuberculata* infestations were correlated with lesser trichome density, highest light reflection and highest chroma in the sprout and the plant’s superior third portion leaves. Low *A. aepim* infestation in both crop seasons made it impossible to verify its correlation with the studied cassava plant characteristics. The cultivars IAPAR 19 and IAC 14 could contribute towards *B. tuberculata* management in regions with a history of whitefly infestation.

## 1. Introduction

The cassava plant (*Manihot esculenta* Crantz) has great importance in human nutrition [1,2]. The world’s largest producers of cassava in decreasing order are Nigeria, Thailand, the Democratic Republic of the Congo, Ghana and Brazil [3]. Pest occurrence can restrain the cassava cultivation, especially in favorable environmental conditions [4,5,6].

Whitefly species are major pests of agricultural crops around the world, from the Americas to Asia and Africa [7,8,9,10]. In cassava cultivations, especially in the Neo tropical regions, whiteflies are serious pests and are responsible for significant economic losses annually [11]. Different species of whiteflies dominate divergent cassava cultivation regions. For instance, in Brazil, *Bemisia tuberculata* (Bondar, 1923) and *Aleurothrixus aepim* (Goeldi, 1886), (Hemiptera: Aleyrodidae), are important pests with occurrences in different cassava-cultivating regions [12,13].

Under favorable development conditions, the whiteflies can cause severe damage, debilitating the cassava plants through sap suction [14]. As a consequence, the plants exhibit chlorosis symptoms and premature leaf fall, and in addition, the honeydew secreted by the insect favors the development of sooty mold (*Capnodium* sp.) which impairs the plant’s photosynthetic capability and gas exchange [15]. Whiteflies are the main pests in cassava-cultivating countries where the transmission of the African Cassava Mosaic Virus (ACMV) greatly impairs productivity, being reduced from 20 to 95% [16,17].

The use of synthetic insecticides is still the main whitefly control strategy [18], however, its indiscriminate use can cause human intoxication risks, environmental imbalance and the selection of insecticide-resistant populations [19,20]. Thus, the development of environmentally sound, resilient and complementary management strategies, such as host plant resistance and biological control agents, should be fostered [20,21,22,23]. The use of resistant cultivars can reduce infestation with pests causing low environmental imbalance, and in addition, in most instances, host plant resistance acts synergistically with biological control in integrated pest management programs [23,24,25].

However, in Brazil, there are a lack of studies to better understand the resistance mechanisms of cassava cultivars against whiteflies. However, previously, multiple studies have indicated the existence of chemical, morphological and/or physical mechanisms in cassava that act against pests [25,26,27,28].

Within the morphological resistance factors, the presence of trichomes and the leaf coloration can influence many insects, attracting or repelling their actions towards the host plants [28,29]. Cultivars with physical characteristics which tarnish the establishment of the whitefly could be recommended for areas with a history of infestation of this pest. However, in field conditions, trichome effects against *B. tuberculata* and *A. aepim* have not been well studied, impairing the development and recommendation of cultivars less susceptible to the whitefly.

Therefore, the objectives of this study were to identify the relationship between whitefly occurrence, and the trichome density and leaf color in different cassava cultivars.

## 2. Materials and Methods

### 2.1. Study Area and Climate

The study was conducted in the city of Londrina (23°21′ S; 51°10′ W; 610 m.a.s.l.), from November 2014 to October 2015 and July 2016 to May 2017, in the experimental station of Instituto de Desenvolvimento Rural do Paraná IAPAR-EMATER (IDR-Paraná).

According to the Köppen classification, the experiment site has a humid subtropical climate (Cfa). The median temperature and accumulated rain during the first and second year were 22.7 °C and 1928.9 mm, and 21.8 °C and 1307.6 mm, respectively (data obtained from the meteorological station of IDR-Paraná, located at 600 m away from the experimental site). 

### 2.2. Experimental Design and Stem-Seed Plantation

The experiment was conducted in a randomized block design, with nine treatments (IAPAR 19, IPR Upira, IPR União, IAC 576-70, IAC 14, IAC 90, Catarina Branca, Santa Helena and Baianinha) and five repetitions (parcels). Each repetitions had an area of 36 m^2^ (6 m × 6 m), with six lines, each line containing six cassava plants. The spacing between lines and between plants was 0.90 m and 0.80 m, respectively. The cultivars were manually planted in two crop seasons, in the early November in 2014 and early July for 2016. No fertilization was performed in both crop seasons. 

### 2.3. Evaluation of B. tuberculata and A. aepim Infestation on Cassava Leaves

The *B. tuberculata* and *A. aepim* nymph population was quantified through the visual inspections of four plants in the middle region of each parcel. The nymphs were differentiated through distinct behavior characteristics, whereas *A. aepim* specimens have gregarious behaviors and wax secretion; the opposite is true for *B. tabaci*. The cassava plants were divided into three regions—lower, middle and superior—and four abaxial leaves’ surfaces, on the superior third of each plant, were inspected, following the North, South, East and West positions. The inspections were performed on the plant’s superior third region due to the whitefly’s oviposition preference in this region [30]. Eleven surveys were conducted from February to October in the first year, in intervals of approximately 20 days, and twenty surveys were performed from November 2016 to May 2017 in the second year, in intervals of approximately 10 days.

### 2.4. Material Preparation and Microscopic Reading for Leaf Trichomes Quantification

Non-expanded leaf (sprout) and the first expanded leaf in the plant’s superior third were gathered 210 days after planting (before flowering), in accordance with Schoonhoven’s observation [31]. 

The material preparation for scanning electron microscope (SEM) followed the methodological proceedings proposed by Zinsou et al. [32]. An area of 4 mm^2^ (2 × 2 mm) from the median section of the central leaf lobe of each cassava cultivar was extracted using a scalpel. The samples were washed in a phosphate-buffered saline solution (0.1 M, pH 7.2). Afterwards, the samples were dehydrated using an ascending concentration of ethanol (30, 50, 70, 90 and 100%) in 10 min intervals. The samples were then critical point-dried in a CO2 atmosphere at 40 °C, gold coted and photographed under an SEM (Phillips FEI, model Quanta 200) with a 400× magnification lens.

After producing the micrographs, the number of trichomes was counted in three randomly allocated areas (0.032 µm^2^). The trichome density per cm^2^ was estimated using these areas’ mean (Appendix A).

### 2.5. Determination of the Colorimetric Parameters 

The colorimetric parameters were performed in field conditions from the same superior third of planta utilized to quantify whitefly nymphs. Five sprouts and the adaxial surface of five leaves per parcel were measured, and the leaves were placed on a black background clipboard to avoid the interference of reflected solar light. 

The colorimetric parameters were obtained through the portable colorimetric device Konica Minolta^®^ (model Chroma Meter CR-400), following the color system of Commision Internationale de L’Éclairage (CIE), which uses the colorimetric parameters luminosity (L*) and chroma (a* and b*), set in standard coordinates within a tridimensional space of color. 

The value of L* ranges from 0 to 100, where 0 is black and 100 is white. The value of a* indicates red when it is represented by positive numbers and green when represented by negative numbers. The value of b* is positive when the object is yellow and negative when it is blue [33]. These parameters correlate consistently the color values with the visual perception [34].

### 2.6. Statistical Analysis 

The normality and homogeneity were verified for the whitefly nymph density per leaf, non-glandular trichome density per cm^2^ and colorimetric parameters. When necessary, the data were transformed using the square root (x + 0.5), where x is the original value. The variance analysis (ANOVA) was performed, and the means compared by the Scott–Knott test (α = 5%).

The variables of trichome density and colorimetric parameters were correlated with the whitefly nymph infestation utilizing the Pearson correlation test (***r***).

## 3. Results

For the 2014/2015 crop season, the lowest *B. tuberculata* infestations were observed on the IAC 14 and IAPAR 19 cultivars, with 5.41 and 3.52 nymphs per leaf, respectively, while the highest *B. tuberculata* infestations occurred on the IPR União, IPR Upira and Baianinha cultivars, with 15.17, 11.85 and 10.72 nymphs per leaf, respectively (Table 1).

Although there was a difference among the cultivars in the 2016/2017 crop season, with highest infestation of *B. tuberculata* in IPR Upira (0.12) and IAC 90 (0.08), these differences should be analyzed with caution, since its value represents less than one nymph every five leaves. The lowest whitefly infestation in the 2016/2017 crop season was probably due to some unfavorable meteorological conditions.

The *A. aepim* infestation was inferior in both crop seasons when compared to the *B. tuberculata*. In the 2014/2015 crop season, the lowest densities of *A. aepim* was observed in IPR União, Baianinha, Catarina Branca, Santa Helena, IAC 576-70, IAC 14, IAPAR 19 with number of nymphs per leaf ranging from 0.66 to 1.99. Due to low infestations, differences between the cassava cultivars for *A. aepim* infestation could not be quantified in the crop season of 2016/2017 (Table 1).

There is a great variation among studied cultivars in the number of non-glandular trichomes on the sprout leaves (Δ = 13055.6) and on the plant’s apical leaves (Δ = 5833.3). The highest non-glandular trichome densities were observed in IPR União, IAC 90 and IPR Upira cultivars, with trichomes density per cm^2^ ranging from 4861.1 to 13055.6 for sprout leaves and ranging from 3472.2 to 5833.3 for the plant’s apical leaves. The IAC 14 genotype had the less trichome density, with 277.8 and 416.7 trichomes/cm^2^ for sprout leaves and plant’s apical leaves, respectively. No presence of non-glandular trichomes were observed on the IAPAR 19 genotype (Table 2).

The association of non-glandular trichome density with the presence of whitefly nymphs was verified in the 2014/2015 crop season. The *B. tuberculata* nymphs were more abundant in cassava cultivars with more trichomes density on the sprout leaves (r = 0.85 and *p* < 0.01), and in plant’s apical leaves (r = 0.81 and *p* = 0.01). The correlation between *A. aepim* nymphs and non-glandular trichome density was not performed, due to the low infestation of whiteflies during both studied years.

Differences among the cultivars to the luminosity (L*) and chroma (b*) parameters were verified in sprout leaves and plant’s apical leaves, while the differences in the chroma a* parameter occcurred only for the sprout leaves. The cultivars IPR Upira, IPR União, Baianinha and IAC 90 presented the lowest values of luminosity and lowest positive values of chroma b* (tendency towards yellow) for the sprouts and plant’s apical leaves (Table 3).

Weak and proportionally-inversed correlations (r = −0.44 to −0.30; *p* < 0.05) were verified between *B. tuberculata* infestation and luminosity, as well as *B. tuberculata* and chroma b* parameters (Table 4), indicating that cassava cultivars with the lowest values of luminosity and lowest positive values of chroma b* (tendency towards yellow) have higher *B. tuberculata* infestations. The correlation between nymph presence and the colorimetric parameters for the *A. aepim* species was not possible to verify, due to its low infestation.

## 4. Discussion

The higher infestation of *B. tuberculata* in the sprout leaves and plant’s apical leaves, is probably a result of the insect’s oviposition preference towards cultivars that are more pubescent. The high density of trichomes might favor the increase of humidity, offering better conditions to the development of nymphs [35]. Many studies have indicated positive correlations between *Bemisia tabaci* (Gennadius, 1889) (Hemiptera: Aleyrodidae) oviposition, and the highly pubescent leaves of the cotton [36,37], butter collard [38], soybean [39], potato [40] and eggplant cultivars [41].

The lowest presence of whiteflies in less pubescent cultivars probably relates to a higher exposition and vulnerability of the nymphs to predation and parasitism. Leaves with high trichomes density impair predator mites [42] and whitefly parasitoids [43] action. Thus, it is probable that non-glandular trichomes on the cassava leaves act as physical barriers to *B. tuberculata* predators and parasitoids.

The relation between pest development and leaf pubescence density is variable. The negative effect of leaves that were more pubescent was verified for *B. tabaci* in cotton [37], as well as for *Aleurotrachelus socialis* Bondar, 1923 (Hemiptera: Aleyrodidae) and the green-mite *Mononychellus tanajoa* (Bondar, 1938) (Acari: Tetranychidae) in cassava [25,27]. However, no relation between trichome density and *B. tabaci* biotype B development was found in cotton and soybean [44,45,46], neither for the mealybug *Phenacoccus manihoti* (Matile-Ferrero) (Hemiptera: Pseudococcidae) nor *A. aepim* in cassava plants [47].

The variation of results observed in the literature indicates that it is necessary to consider which studied pest and host plant (species and variety) is involved, and that the results can vary depending on the studies object. Beyond the trichome density, other factors such as leaf coloration could be involved in the manifestation of pest resistance to certain cultivars.

Insect’s host plant selection is initially related to visual stimuli based on colors [46]. Generally, whitefly species are attracted to the yellow color [48,49,50,51]. Our study corroborates previous studies, which indicate more whitefly attractiveness from surfaces with a reflectance between green and yellow (wavelength 520–600 nm) [52], while the violet light (wavelength 400 nm) influenced their migration habits [53].

A study conducted with *Trialeurodes vaporariorum* Westwood, 1856 (Hemiptera: Aleyrodidae), demonstrated that these insects were very attracted to the “pure yellow” color. However, the spectral transition between green and yellow can also attract or repel them, depending on the luminosity [54]. Indicating that not just the color, but also the luminosity interferes in the attraction or repellence of whiteflies.

Our results for luminosity and chroma b* corroborate with those observed with *B. tabaci* on cotton, in which the adult insects had a lesser preference towards permanency and oviposition on leaves with a high light intensity (L*) and tendency towards a yellow color (positive b* values) [37]. Similar results were found with *B. tabaci* in eggplant, in which, a higher brightness in the evaluated colors (red, green and blue), resulted in a shorter permanency and oviposition of adult insects [40].

A study conducted with cassava cultivars verified that the whitefly *A. aepim* had less preference for leaves with low reflected light intensity, with no difference found for the chroma b* [47], differing from the present study. However, the obtained results in the mentioned study were observed in other cassava cultivars, grown in a greenhouse, while the results of the present study were obtained in field conditions, with different lighting conditions, that may have contributed to the observed difference between the results.

The colors blue, green and red were reported as less attractive to *T. vaporariorum* adults [54]. Our results were similar since there was no correlation between chroma a* (tendency towards green) and the *B. tuberculata* nymph presence was verified.

The plant’s resistance to whitefly is not defined by one single factor, but from an intrinsic and extrinsic interaction with the plant [43]. There are other factors besides trichome density and leaf color that influence whitefly occurrence. Thus, the evaluation of plant-derived volatile organic compounds (VOCs), which play pivotal roles in interactions between host plant and insect herbivores, should also be considered in further studies, aiming towards the development of whitefly-resistant cassava cultivars.

## 5. Conclusions

The lesser trichome density, higher light reflectance (L*) and higher chroma value (b*) could contribute to lesser *B. tuberculata* infestation in cassava cultivars.

The cultivars IAPAR 19 and IAC 14 could contribute to the *B. tuberculata* management in areas with this pest’s infestation history, as well as serving as a resistance genetic source for the development of new cultivars.

## Figures and Tables

**Table 1 insects-14-00004-t001:** Nymph density (median ± standard deviation) of *Bemisia tuberculata* and *Aleurothrixus aepim* per leaf on the plant’s superior third portion, from nine cassava cultivars, Londrina, Paraná, crop season 2014/15 and 2016/17.

Cultivars	*Bemisia tuberculata*				*Aleurothrixus aepim*		
	2014/2015		2016/2017		2014/2015		2016/2017
IPR União	15.17 ± 0.53	a ^1^	0.03 ± 0.01	c	1.99 ± 0.77	b	0.021 ^ns^
IPR Upira	11.85 ± 0.82	b	0.12 ± 0.02	a	3.47 ± 0.90	a	0.014
Baianinha	10.72 ± 0.73	b	0.04 ± 0.01	c	1.38 ± 0.79	b	0.025
IAC 90	8.43 ± 1.21	c	0.08 ± 0.02	b	4.04 ± 1.28	a	0.003
Catarina Branca	8.43 ± 1.27	c	0.04 ± 0.01	c	1.76 ± 0.72	b	0.004
Santa Helena	7.42 ± 0.94	c	0.03 ± 0.01	c	0.76 ± 0.25	b	0.011
IAC 576-70	6.82 ± 0.73	c	0.06 ± 0.01	c	1.10 ± 0.35	b	0.001
IAC 14	5.41 ± 0.54	d	0.04 ± 0.01	c	0.66 ± 0.14	b	0.004
IAPAR 19	3.52 ± 0.59	d	0.02 ± 0.01	c	0.72 ± 0.39	b	0.006
DF Error	32		32		32		-
CV (%)	20.23		40.83		78.27		-
*p*-value	<0.01		<0.01		<0.01		-

^1^ Medians followed by the same letters in the columns do not differ from each other by statistical test Scott–Knott (α = 5%). ^ns^ = no significance.

**Table 2 insects-14-00004-t002:** Estimated non-glandular trichome density (median ± standard deviation) on sprout leaves and leaves from the plant’s superior third portion, in an area of 1 cm², from nine cassava cultivars. Londrina, Paraná.

Cultivars	Estimated Trichomes/cm^2^					
	Sprout Leaves			Plant’s Apical Leaves ^2^		
IPR União	13055.6 ±	636.5	a ^1^	5833.3 ±	1666.7	a
IAC 90	6250.0 ±	416.7	b	5138.9 ±	1683.9	a
IPR Upira	4861.1 ±	636.5	b	3472.2 ±	1048.6	a
Santa Helena	3888.9 ±	1969.1	c	1805.6 ±	636.5	b
Catarina Branca	3194.4 ±	636.5	c	1388.9 ±	962.3	b
IAC 576-70	3055.6 ±	636.5	c	1111.1 ±	636.5	b
Baianinha	2638.9 ±	240.6	c	2083.3 ±	721.7	b
IAC 14	277.8 ±	481.1	d	416.7 ±	0.0	c
IAPAR 19	0.0 ±	0.0	d	0.0 ±	0.0	d
DF Error	18			18		
CV (%)	19.68			20.81		
*p*-value	<0.01			<0.01		

^1^ Medians followed by the same letters in the columns do not differ from each other by statistical test Scott–Knott (α = 5%). ^2^ Transformed values by square root (x + 0.5), where x is the original value.

**Table 3 insects-14-00004-t003:** Median values (±standard deviation) from the Luminosity (L*) and Chroma (a* and b*) parameters in sprout leaves and plant’s apical leaves, from nine cassava cultivars. Londrina. Paraná.

Cultivars	Sprout Leaves			Plant’s Apical Leaves		
	L*	a*	b*	L*	a*	b*
IAC 576-70	36.2 ± 1.3 a ^1^	−12.1 ± 1.5 c	21.0 ± 2.1 a	38.5 ± 1.2 a	−16.3 ± 0.6	23.3 ± 1.1 a
Catarina Branca	34.9 ± 1.8 b	−10.0 ± 2.8 c	18.8 ± 4.1 a	37.8 ± 0.5 a	−14.8 ± 2.8	22.7 ± 0.9 a
Santa Helena	34.5 ± 1.6 b	−7.9 ± 3.0 b	19.5 ± 2.9 a	37.1 ± 0.9 a	−15.4 ± 0.7	21.8 ± 1.2 a
IAPAR 19	34.1 ± 1.2 b	−5.9 ± 1.6 b	16.3 ± 2.1 b	37.5 ± 0.4 a	−15.4 ± 0.7	22.2 ± 0.5 a
Baianinha	33.3 ± 1.4 c	−7.3 ± 1.6 b	14.5 ± 2.1 b	35.3 ± 0.9 b	−14.3 ± 1.2	18.0 ± 1.3 b
IPR União	33.1 ± 0.7 c	−6.9 ± 2.0 b	13.5 ± 2.0 b	35.8 ± 0.7 b	−14.7 ± 0.8	19.6 ± 0.7 b
IAC 14	32.3 ± 0.8 d	−4.8 ± 1.7 a	12.0 ± 1.8 c	37.2 ± 0.9 a	−14.5 ± 0.4	18.9 ± 1.2 b
IAC 90	31.8 ± 1.7 d	−6.9 ± 2.8 b	14.9 ± 2.9 b	35.0 ± 0.8 b	−12.8 ± 2.6	18.4 ± 1.0 b
IPR Upira	30.1 ± 1.3 e	−2.7 ± 2.1 a	11.2 ± 2.3 c	35.1 ± 0.7 b	−14.1 ± 0.6	19.3 ± 1.0 b
DF Error	32	32	32	32	32	32
CV (%)	2.78	29.51	13.97	2.31	9.54	5.09
*p*-value	<0.01	<0.01	<0.01	<0.01	ns	<0.01

^1^ Medians followed by the same letters in the columns do not differ as shown by the statistical test Scott–Knott (α = 5%). ns. = no significance.

**Table 4 insects-14-00004-t004:** Pearson correlation test (r) values between the colorimetric parameters (L*, a* and b*) and the *B. tuberculata* nymph density present in sprout leaves and superior third portion leaves, from nine cassava cultivars. Londrina. Paraná.

	Sprout Leaves			Plant’s Apical Leaves		
	L*	a*	b*	L*	a*	b*
r (Pearson)	−0.31	0.10	−0.31	−0.44	0.04	−0.30
*p*-value	0.04	0.53	0.04	<0.01	0.81	0.04

## Data Availability

The data presented in this study are available on request from the corresponding author. The data are not publicly available due to the Instituto de Desenvolvimento Rural do Paraná IAPAR-EMATER’s policies.

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
