# Peer review of "The Leaf Color and Trichome Density Influence the Whitefly Infestation in Different Cassava Cultivars"

_insects, 2022, doi:10.3390/insects14010004_

Round 1

Reviewer 1 Report (New Reviewer)

The manuscript tries to associate the trichomes from cassava to the plant’s tolerance to whiteflies. The results point those plants with less trichomes, high light reflection and chroma, are less prone to whiteflies attack. 

Major comments that should be addressed by authors are as follows:

1.     It is not very clear the methodology used to evaluate the whiteflies infestation. As the experiment was done in the field, I assume the infestation was natural. How the authors did the differentiation between the two species during the analysis? 

2.     Did the nymph’s number was the only parameter evaluate? Why? Because in other species, the trichomes can affect other phases of whitefly’s lifecycle. 

3.     I really miss a better explanation about trichomes in the species. The authors did SEM images, but I didn’t find them. In cassava, are there just non-glandular trichomes? 

4.     As the experiment was done in the field, how the authors can make sure that just trichomes and leaf color parameters affect the whiteflies behavior? The authors points at the end of manuscript that other factors could interfere on the experiment, like the plant chemical compounds. And in my opinion, this is the data that can make this work much more complete. 

Minor comments: 

1.     I suggest the inclusion of more keywords like: trichomes, herbivory, cassava. 

2.     In other species, like S. lycopersicum, the major concern with whiteflies is because they are viruses’ vectors. Is it a problem also in cassava? This point could be exploring a little bit on the introduction. 

Author Response

Major comments:

  1. It is not very clear the methodology used to evaluate the whiteflies infestation. As the experiment was done in the field, I assume the infestation was natural. How the authors did the differentiation between the two species during the analysis?
    1. The distinguishability between nymphs is quite simple, as both specimens demonstrate different behaviors and phenology, Aleurotrixus aepim nymphs release wax with a woolly aspect on their dorsal region and demonstrate gregarious behavior, while Bemisia tuberculata does not produce wax and does not demonstrate gregarious behavior. You may find a more complete explanation in the following manuscript: https://doi.org/10.5433/1679-0359.2022v43n1p311
  2. Did the nymph’s number was the only parameter evaluate? Why? Because in other species, the trichomes can affect other phases of whitefly’s lifecycle.
    1. Only nymphs were evaluated due to two factors. First, that adults can relocate from leaf to leaf and plant to plant, thus their behavior could not be accurately evaluated in the given sample conditions, neither in the given field conditions. To evaluate adults, the study should be performed in controlled environments. Second, leaf characteristics, like color and trichome density, are proven to affect the insect’s host plant selection, be it for feeding or reproduction. Thus, one way to evaluate host plant preference is to count eggs or juvenile stages present in the given plant tissue, as they demonstrate measurable variables. Counting eggs would be inexecutable in filed conditions, given their reduced size. References: https://doi.org/10.1007/s13355-013-0219-x; https://doi.org/10.1093/ee/26.5.1049; https://doi.org/10.1111/j.1440-6055.2010.00796.x; https://doi.org/10.1590/S1519-566X2007000300013; https://doi.org/10.1146/annurev.ento.53.032107.122456; https://doi.org/10.1371/journal.pone.0153880.
  3. I really miss a better explanation about trichomes in the species. The authors did SEM images, but I didn’t find them. In cassava, are there just non-glandular trichomes?
    1. The SEM pictures will be available as supplementary material as requested. Only non-glandular trichomes are present on the cassava leaf’s surface.
  4. As the experiment was done in the field, how the authors can make sure that just trichomes and leaf color parameters affect the whiteflies behavior? The authors points at the end of manuscript that other factors could interfere on the experiment, like the plant chemical compounds. And in my opinion, this is the data that can make this work much more complete.
    1. We recognize that chemical compounds influence insect occurrence, through antibiosis, and that complementary studies should evaluate these interactions. However, chemical composts were not evaluated in the given study. Furthermore, many articles indicate the trichome density influence on insect occurrence, be it from antixenosis (oviposition), or protection against natural enemies.

Minor comments:

  1. I suggest the inclusion of more keywords like: trichomes, herbivory, cassava
    1. We feel that these new keywords would be redundant, as they’re found in the article’s title. The keyword herbivory may indicate a study focused on insect feeding or plant damage, which were not evaluated in the present study.
  2. In other species, like lycopersicum, the major concern with whiteflies is because they are viruses’ vectors. Is it a problem also in cassava? This point could be exploring a little bit on the introduction.
    1. Yes, cassava also suffers from whitefly virus transmission. The crop’s most important disease, the African Cassava Mosaic Virus (ACMV), is transmitted by the Bemisia tabaci We agree with the given suggestion, a new sentence was added in the introduction section. Sentence: Whiteflies are main pests in cassava cultivating countries where the transmission of the African Cassava Mosaic Virus (ACMV), greatly impairs productivity, being reduced from 20 to 95% [https://doi.org/10.1094/PD-74-0404; https://doi.org/10.1111/j.1439-0434.1990.tb01179.x].

Reviewer 2 Report (New Reviewer)

The manuscript by Postoria et al. titled, “The leaf color and trichome density influence the whitefly occurrence in different cassava cultivars” describes the whitefly species’ interactions with several cassava cultivars. The study was conducted under field conditions over two growing seasons.  Pastoria et al report cassava cultivars with fewer trichomes and higher light reflection from the leaf surface had reduced whitefly colonization. The manuscript has good scientific information with implications for cassava breeding programs. However, the manuscript does not describe the methodology used for field identification of whitefly species, which is very imperative to draw a conclusion. Other than this, I have minor editorial comments in the attached pdf version of the manuscript. 

Author Response

Direct editing on the paper

Dear reviewer,

We found that pretty much every suggestion to the text was relevant, thus the corrections were made, and the extra information and citations suggested in the introduction were added.

However, the last suggestion, to be included at the end of the discussion, was not agreed by the authors as the present study did not evaluate virus transmission or the plant’s response towards virus infection, thus a conclusion in this manner may obscure our previously discussed results.

The changes in the manuscript were recorded using the review function of Microsoft Word, as requested by the journal.

Round 2

Reviewer 1 Report (New Reviewer)

The authors answered my initial questions, but one thing still intrigued me. Taking into account the number of trichomes presented in the results, I expected to see images of the União cultivar looking much hairier than what was shown in the supplemental pictures. From the SEM images presented, it seems that all cultivars are glabrous. Therefore, the trichomes would not be responsible for the change in the color parameters of the leaves.

Author Response

Reviewer 1

Dear reviewer one,

The only glabrous leaf we found was from the IAPAR 19 cultivar, followed by the IAC 14 cultivar, which demonstrated the fewest trichomes, not significantly different from IAPAR 19. Both these cultivars also demonstrated the lowest nymph densities, that correlated to the leaf’s color characteristics, like luminosity and yellow chroma. The assumption that trichomes directly affect the leaf’s color characteristics was not made in the present study, we only assumed that these characteristics affect insect attraction and repelling, since the trichomes present little to no color and are mostly found in the abaxial region of the leaves.

Our conclusion was “The lesser trichome density, higher light reflectance (L*) and higher chroma value (b*) could contribute to lesser B. tuberculata infestation in cassava cultivars”, thus the combination of these characteristics results in fewer nymph densities, not that trichome densities induce color characteristics changes.

Reviewer 2 Report (New Reviewer)

Here are few minor (not compulsory though) suggestions.

Connected lines (62-72) might become more coherent (just a suggestion below, I mentioned this in the first review too). I agree with the authors about not talking about viruses later in the manuscript. However, that sentence does not look very attractive, I tried changing it a bit. Authors can decide how they want it. Other than this, the manuscript looks good and is ready to fly.

Line 62: Since a single interbreeding population of whitefly can dominate the whole farmscape, resistant management is particularly important in areawide whitefly management programs [https://doi.org/10.3390/insects11120834].  Thus, the development of environmentally sound, resilient, and complementary management strategies such as host plant resistance and biological control agents needs to be been fostered [20-23].  The use of resistant cultivars can reduce infestation with pests causing low environmental imbalance, in addition, in most instances, host plant resistance acts synergistically with biological control in integrated pest management programs [23-25]. However, in Brazil, there is a lack of studies to better understand the resistance mechanisms of cassava cultivars against whiteflies. Although, previously multiple studies have indicated the existence of chemical, morphological, and/or physical mechanisms, in cassava, that act against pests [25-28].

Line 269: Thus, the evaluation of plant-derived volatile organic compounds (VOCs) which play pivotal roles in interactions between host plants with insect herbivores should also be considered in further studies directed toward the development of whitefly-resistant cassava cultivars. 

Author Response

Reviewer 2

Dear reviewer two,

The suggestion from line 62 was not accepted as our study aimed to quantify two specimens, which dominate the cassava cultivation landscape in Paraná State, Bemisia tuberculata and Aleurothrixus aepim. Commenting that a single interbreeding population exists would contradict our initial study parameters. Unfortunately, the A. aepim specimen was scarce in both crop seasons, but it was important to relay its presence in the evaluated region. We believe that further studies comprised of samples from different regions in the country are necessary to better understand the whitefly specimens’ distribution. Thus, we do not want to assume that a single interbreeding population exists in the given region.

We believe that the sentence in line 62 “The use of synthetic insecticides is still the main whitefly control strategy [18], however, its indiscriminate use can cause human intoxication risks, environmental im-balance and selection of insecticide resistant populations [19,20]”, correctly depicts the current scenario for whitefly control and its difficulties.

The suggestions for lines 63-79 were implemented in the text as requested.

We changed the sentence in line 270, as we believe you presented a more robust and descriptive suggestion for further studies.

Round 3

Reviewer 1 Report (New Reviewer)

Still, taking into account the number of trichomes in the IMAGES obtained by the authors, where all genotypes have very low densities of trichomes, I do not believe that the low density of trichomes could be impacting the results with whiteflies. The lower presence of nymphs would result from other factors not explored in work.

Author Response

Reviewer 1

Dear reviewer one,

Firstly, we would like to apologize, for we had only sent the SEM images from the superior third portion leaves. Annexed in the supplementary files are now the SEM images from the shoot and superior third portion leaves. We hope you may now clearly visualize the differences between the cultivar’s trichome patterns and densities.

Secondly, the number of trichomes was counted in three randomly allocated areas of the SEM images, on a surface of 0.032 µm². Then, the trichome density per cm² was estimated using these three areas’ mean trichomes.

We believe our results were supported by our findings, be it, the nymph density manually counted on field, the leaf colorimetric characteristics and of course, the estimated trichome density per cm² of leave area. All data were thoroughly statistically analyzed, and patterns were found. Thus, our conclusion that lesser trichome densities, compiled with the leaf color characteristics of higher light reflectance (L*) and greater chroma (b*) values, negatively affects whitefly nymph density, is statistically supported and discussed with other research papers, which demonstrate similar results.

We have changed the Table 2’s title to accurately represent the estimated results found in the given area.

If you believe the nymph densities were affected strictly by other factors, not researched in the given paper, we can only agree that there are many other factors also affecting the whiteflies’ preference, but we also believe our research is a step towards better understanding these relationships (plant-insect), and we agree that further research is needed, analyzing VOCs, plant genetic characteristics, whitefly ecology, also further plant chemical, physical and biological barriers towards pests.

This manuscript is a resubmission of an earlier submission. The following is a list of the peer review reports and author responses from that submission.

Round 1

Reviewer 1 Report

This manuscript studied the population difference of two species of whitefly in different cassava varieties. The correlation between the the whitefly nymph quantification and characteristics of cassava leaves of different varieties, including trichome density, luminosity and chroma was analysed. This study provided reference for screening pest-resistant cassava varieties and related resistance genes of cassava.

There are several questions to be discussed here.

1.    We know that whiteflies are very tiny and similar that it is difficult to find morphological differences even under a microscope. This paper determined the whitefly species through field investigation. Could you explain how to distinguish the nymphs of two species by visual observation in the field?

2.    Does the analysis of Aleurotrixus aepim, a particularly low population of whitefly, reliably support the conclusion?

3.    The paper introduced that the whiteflies were investigated several times each year, but it did not explain how to design the survey time. Is the result the average of all times? The population of whiteflies varies greatly in different seasons and at different times. It would be interesting to see if there was any analysis of changes in the population of whiteflies over time.

4.    We noted that there may also be differences in trichome density, luminosity and chroma in different parts of the cassava leaves (sprout leaves and plant’s apical leaves). It might also be interesting to consider the differences in whiteflies in different parts of the same cassava.

Minor revision

In table 4, ” p-valor ” should be a spelling mistake.

the format of references, such as italics of scientific names.

Author Response

  1. We know that whiteflies are very tiny and similar that it is difficult to find morphological differences even under a microscope. This paper determined the whitefly species through field investigation. Could you explain how to distinguish the nymphs of two species by visual observation in the field?

A1: The distinguishability between nymphs is quite simple, as both specimens demonstrate different behaviors and phenology, Aleurotrixus aepim nymphs release wax with a woolly aspect on their dorsal region and demonstrate gregarious behavior, while Bemisia tuberculata does not produce wax and does not demonstrate gregarious behavior. You may find a more complete explanation in the following manuscript: https://doi.org/10.5433/1679-0359.2022v43n1p311

  1. Does the analysis of Aleurotrixus aepim, a particularly low population of whitefly, reliably support the conclusion?

A2: The conclusions were based on Bemisia tuberculata, as there were statistical results to enforce the findings. Due to a low population of Aleurotrixus aepim, no conclusions were made about this specimen, only discussed the results found during the field study.

  1. The paper introduced that the whiteflies were investigated several times each year, but it did not explain how to design the survey time. Is the result the average of all times? The population of whiteflies varies greatly in different seasons and at different times. It would be interesting to see if there was any analysis of changes in the population of whiteflies over time.

A3: The results shown for both crop seasons are the averages as described in section 2.3 of material and methods, an average of 11 evaluations from the first season and 20 evaluations from the second season. The scopus of the present study was to determine the relation between whitefly occurrence and plant traits, such as trichome density and leaf color in relation to the nine evaluated cultivars. Understanding that an insect’s population dynamic is influenced by environmental conditions (exp: temperature, precipitation and neighboring host plants), a conclusion on insect population distribution, based on solely two seasons, as well as, on nine cultivars, would be very mediocre. An ideal situation would be to eliminate the cultivar variable and repeat the analyses in various seasons, thus allowing a correlations between insect population dynamics and the environmental conditions.

  1. We noted that there may also be differences in trichome density, luminosity and chroma in different parts of the cassava leaves (sprout leaves and plant’s apical leaves). It might also be interesting to consider the differences in whiteflies in different parts of the same cassava.

 A4: Since whiteflies mainly oviposit their eggs in the plant’s superior third, a higher emphasis was attributed to this region. As no oviposition occurs on sprout leaves, there is no point in correlating this specific region’s characteristics with whitefly occurrence. Also, before the execution of the present study, a pre-analyses revealed a higher nymph concentration on the superior third of the plants.

Minor revision: In table 4, ” p-valor ” should be a spelling mistake and the format of references, such as italics of scientific names.

A6: These minor revisions have been attended to.

Reviewer 2 Report

In the manuscript “The leaf color and trichome density influence the whitefly occurrence in different cassava cultivars” by Pastório and co-authors the authors try to find the relation between Bemisia tuberculata and Aleurotrixus aepim occurrence and leaves color and pubescence in different cassava cultivars. The subject of the study is another contribution to general understanding of the influence of morphological characteristics of leaves on pests occurrence. Such studies are important in developing non-chemical pest control methods.

In general the work is well structured and presented. Key words ‘Chroma Meter’ and ‘Konica-Minolta’ should be replaced e.g. ‘Aleurotrixus aepim’, ‘colorimetric parameters’. The introduction is clearly written and no inappropriate self-citations by authors are present. It explained major information for understanding the other sections of this manuscript. The experimental plan has been mostly properly explained and methods adequately presented. However, the description of some parts of methodology should be more extensive. Please explain L 92-93: ‘The inspections were performed on the plant’s superior third due to the whitefly’s oviposition preference in this region.’ and L 100-101: ‘… methodological proceedings proposed by Zinsou et al.’ In its current form, this section does not contain enough information which will make it impossible to repeat the tests performed by the authors.

Results unfortunately are narrow, especially concerning insects. In Table 4  ‘p-valor’ correct to ‘p-value’.

In my opinion the results are to limited to draw correct conclusions.

Author Response

  1. In general the work is well structured and presented. Key words ‘Chroma Meter’ and ‘Konica-Minolta’ should be replaced e.g. ‘Aleurotrixus aepim’, ‘colorimetric parameters’. The introduction is clearly written and no inappropriate self-citations by authors are present. It explained major information for understanding the other sections of this manuscript. The experimental plan has been mostly properly explained and methods adequately presented. However, the description of some parts of methodology should be more extensive. Please explain L 92-93: ‘The inspections were performed on the plant’s superior third due to the whitefly’s oviposition preference in this region.’ and L 100-101: ‘… methodological proceedings proposed by Zinsou et al.’ In its current form, this section does not contain enough information which will make it impossible to repeat the tests performed by the authors.

A1: The Keywords ‘Chroma meter’ and ‘Konica-Minolta’ were substituted with ‘Leaf colorimetric parameters’ as suggested. However, the suggestion to include the keyword ‘Aleurotrixus aepim’ was not accepted, as this specimen’s occurrence was low, thus making a conclusion unreasonable, while misguiding potential readers to the results they may find in the paper. Further details were included in the material and methods section as suggested.

2. Minor revision: In Table 4  ‘p-valor’ correct to ‘p-value’.

A2: The recommended suggestion was followed.

3. Results unfortunately are narrow, especially concerning insects. In my opinion the results are to limited to draw correct conclusions

A3: Although the results are narrow, statistical differences were verified, exploring the data using the best available methods. We also agree that further studies should be conducted to achieve a more robust conclusion. Our conclusion was altered to indicate possibility of less B. tuberculata infestations in less trichome dense, higher reflectance and higher chroma value cultivars.